# Temporal Trends in Renal Replacement Therapy in Community-Based People with or without Type 2 Diabetes: The Fremantle Diabetes Study

**DOI:** 10.3390/jcm11030695

**Published:** 2022-01-28

**Authors:** Wendy A. Davis, Aron Chakera, Edward Gregg, Daniel McAullay, Timothy M. E. Davis

**Affiliations:** 1Medical School, The University of Western Australia, Crawley, WA 6009, Australia; wendy.davis@uwa.edu.au (W.A.D.); aron.chakera@uwa.edu.au (A.C.); 2Department of Renal Medicine, Sir Charles Gairdner Hospital, Nedlands, WA 6009, Australia; 3School of Public Health, Imperial College London, London SW7 2BX, UK; e.gregg@imperial.ac.uk; 4Kurongkurl Katitjin Centre for Indigenous Australian Education and Research, Edith Cowan University, Mount Lawley, WA 6050, Australia; mcaullay@gmail.com

**Keywords:** type 2 diabetes, renal replacement therapy, incidence, Aboriginal

## Abstract

Background: Although rates of cardiovascular disease complicating type 2 diabetes are declining, equivalent data for renal replacement therapy (RRT) are conflicting. The aim of this study was to characterize temporal changes in RRT incidence rates (IRs) in Australians with or without type 2 diabetes. Methods: Participants with type 2 diabetes from the Fremantle Diabetes Study Phases I (FDS1; *n* = 1291 recruited 1993–1996) and II (FDS2; *n* = 1509 recruited 2008–2011) were age-, sex- and postcode-matched 1:4 to people without diabetes and followed for first hospitalization for/with RRT. Five-year IRs, IR ratios (IRRs) for those with versus without diabetes in FDS1 and FDS2, and IR differences (IRDs), were calculated. Results: The 13,995 participants had a mean age of 64.8 years and 50.4% were males. For the type 2 diabetes cohorts, the 5-year RRT IR was nearly threefold higher in FDS2 versus FDS1 (IRR (95% CI): 2.85 (1.01–9.87)). Sixteen more participants with type 2 diabetes/10,000 person-years received RRT in FDS2 than FDS1 compared with an IRD of 2/10,000 person-years in those without diabetes. Type 2 diabetes increased RRT risk at least 5-fold. This increased risk was greater in Aboriginal participants who were relatively young when RRT was initiated and more prone to rapid progression to RRT. Multivariable analysis using the combined FDS type 2 diabetes cohorts confirmed albuminuria as a strong independent RRT risk factor. Conclusions: The incidence of RRT is increasing substantially in Australians with type 2 diabetes, especially in Aboriginals who progress to RRT more rapidly at a younger age than non-Aboriginals.

## 1. Introduction

Several longitudinal studies from North America, Europe, Asia and Australia have demonstrated that there has been a decline in chronic macrovascular complications of diabetes over the past few decades [1,2,3,4] as well as in all-cause and cardiovascular disease (CVD) death [5,6], reflecting improving CVD risk factor management [7,8,9]. Although these encouraging trends may be stabilizing and even reversing based on even more recent US data [10,11], there is still the possibility that other complications such as chronic kidney disease (CKD) may become more prominent as longevity increases [12].

Although there is some evidence of CKD leading to end-stage kidney disease (ESKD) and the need for renal replacement therapy (RRT) has been declining in parallel with CVD in type 2 diabetes in some countries, the trend has not been as marked as that for major CVD outcomes such as myocardial infarction (MI) and stroke [2,3]. In addition, recent national data relating to temporal trends in CKD complicating type 2 diabetes are inconsistent, with some studies showing a reduction [3,13] and others an increase [12,14,15,16,17] in CKD hospitalizations and progression to RRT. The use of administrative data in these studies has well-recognized limitations, including incomplete ascertainment of diabetes and its type, misclassification of endpoints, inconsistent follow-up and missing patient-level data, and lack of detailed data relating to risk factors and clinical management. In addition, the decision to start RRT may depend on local eligibility criteria, as well as the availability of specialist renal services, dialysis facilities, and donor kidneys. These factors, which may disadvantage people with diabetes and associated comorbidities such as obesity and CVD [18,19,20], are likely to differ across countries and change with time. An important additional consideration in access to and uptake of RRT remains ethnic/racial background. Some racial groups such as Black Americans and Asians have both an increased risk of type 2 diabetes and an increased risk of progression to CKD and ESKD than Whites, but are less likely to receive RRT [21,22,23]. The burden of ERSD is also higher in Aboriginal than White Australians [24], but a recent national database study acknowledged that registration of more severe cases might lead to an overestimation of rates of RRT complicating type 2 diabetes in Aboriginal Australians [14].

We have previously shown that CVD outcomes in well-characterized Australians with type 2 diabetes from the Fremantle Diabetes Study Phases I (FDS1; recruited 1993–1996) and II (FDS2; recruited 2008–2011) improved significantly between phases and proportionately more than in matched people without diabetes from the same geographic area [1]. The aims of the present study were (i) to determine whether changes in five-year incidence rates (IRs) of RRT paralleled those of CVD outcomes in the 15 years separating the two FDS phases, (ii) to compare these RRT IRs with those from matched people without known diabetes from the same geographical area, and (iii) to examine the independent predictors of RRT within the pooled FDS type 2 diabetes cohorts including the influence of FDS phase. We also analyzed our data according to whether or not participants were Aboriginal to quantify the relative burden of RRT in these participant groups.

## 2. Patients and Methods

### 2.1. Study Site, Participants, and Approvals

The FDS1 is an observational, longitudinal study of known diabetes conducted in a postcode-defined geographic area surrounding the port city of Fremantle in the Australian state of Western Australia (WA) [25]. Recruitment took place between 1993 and 1996, with follow-up for initiation of RRT in the present sub-study to end-2017. The FDS2 utilized the same design as FDS1 with recruitment between 2008 and 2011 and, in the case of the present study, follow-up to end-2016. Socio-economic data from the study catchment area at the time of FDS2 recruitment showed an average Index of Relative Socio-economic Advantage and Disadvantage [26] of 1033, with a range across the included postcodes of 977–1113. These figures parallel the Australian national mean ± SD (1000 ± 100), confirming that FDS participants were from a representative urban Australian community.

Participants in both FDS phases were identified from hospital, clinic, and primary care patient lists, advertising through local print media, pharmacies, optometrists, networks of health care professionals, and, in the case of FDS2, third-party mail-outs to registrants of the Australian National Diabetes Services Scheme and the National Diabetes Register [25]. Details of recruitment, sample characteristics including classification of diabetes types, and non-recruited patients have been published previously [25,27]. The FDS1 protocol was approved by the Fremantle Hospital Human Rights Committee in February 1993, and the FDS2 protocol by the Human Research Ethics Committee of the Southern Metropolitan Area Health Service in October 2007 (07/397). All participants gave written informed consent.

In FDS1, 2258 people with diabetes were identified from a population of approximately 120,000, and 1426 (63%) were recruited, of whom 1296 (91%) had clinically defined type 2 diabetes. In FDS2, 4639 people with diabetes were identified from a population of approximately 157,000, and 1668 (36%) were recruited, of whom 1509 (90%) had type 2 diabetes. Four age-, sex- and postcode-matched residents without any prior documentation of diabetes in health databases were randomly selected from the study catchment area for each FDS1 and FDS2 participant at the time of their enrolment using the WA Electoral Roll of all adults resident in the FDS catchment area and, for FDS2, the WA Registry for Births, Deaths and Marriages. Five of these residents died just before their matched participant with diabetes was enrolled in FDS1 and were therefore excluded. Matches could not be made for five young and four elderly FDS1 participants who were also excluded. This left 1291 FDS1 participants with type 2 diabetes (99.6%) who were matched with 5159 residents without diabetes. In the case of FDS2, the 1509 participants with type 2 diabetes were age-, sex-, and postcode-matched with 6036 residents without diabetes.

### 2.2. Baseline and Annual Assessments

In both FDS phases, assessment at entry and at each annual (FDS1) or biennial (FDS2) review included a comprehensive questionnaire, physical examination, and fasting biochemical tests performed in a single nationally accredited laboratory [25]. Demographic, socio-economic, and lifestyle data were recorded in addition to details of all medical conditions. Ethnic background was based on self-selection, country/countries of birth and parents’ birth and, (in FDS2) country of grandparents’ birth, as well as language(s) spoken at home. Six categories were used, specifically, Anglo-Celt, Southern European, Other European, Asian, Aboriginal Australian, or Mixed/other [25]. Consistent with Australian legal rulings and other studies of Aboriginal Australians with diabetes, we used self-identification and acceptance by the local community as primary criteria for Aboriginality [28]. There were no FDS participants who identified themselves as from a Torres Strait Islander racial background. Patients were requested to bring all prescribed, over-the-counter, and complementary medications to each visit and full details of these were recorded. In FDS2, comprehensive postal questionnaires were sent to participants in the alternate years between face-to-face assessments.

Complications were identified under standard definitions [29]. Albuminuria was assessed from early morning spot urine albumin–creatinine ratio (ACR) measurement and renal impairment from the estimated glomerular filtration rate (eGFR) [30]. Peripheral sensory neuropathy (PSN) was ascertained using the clinical portion of the Michigan Neuropathy Screening Instrument [31]. Retinopathy was defined as one microaneurysm in either eye or worse and/or evidence of laser treatment on direct/indirect ophthalmoscopy (FDS1) or fundus photography (FDS2), and/or external assessment by an ophthalmologist. Patients classified as having prevalent coronary heart disease (CHD) had history of MI, angina, coronary artery bypass grafting, or angioplasty, and those with prevalent cerebrovascular disease had a history of stroke and/or transient ischemic attack. Peripheral arterial disease (PAD) was defined as an ankle brachial index ≤0.90 or a diabetes-related lower extremity amputation.

### 2.3. Ascertainment of Incident Renal Replacement Therapy

Outcomes of interest during follow-up were the first hospital admission for/with RRT, death or five years, whichever came first. The Hospital Morbidity Data Collection (HMDC) documents all public/private hospitalizations in WA since 1970, while the Death Register contains information on all deaths in the state [32]. Both FDS phases have been linked confidentially to these databases through the WA Data Linkage System (WADLS), as approved by the WA Department of Health Human Research Ethics Committee. This source provided validated data on incident events to end-2017 for FDS1 and end-2016 for FDS2. Relevant International Classification of Disease (ICD)-9-CM and ICD-10-AM codes were used to identify RRT in the HMDC. These comprised diagnosis codes V45.1, V56 and 996.81 (ICD-9-CM) and Z49, Z94.0, Z99.2 and T86.1 (ICD-10-AM), and procedure codes 55.69 (ICD-9-CM) and 36503-00 and 36503-01 (ICD-10-AM).

The HMDC was used as a source of additional data to those obtained through individual FDS assessments relating to prevalent/prior disease during the five years prior to study entry, as well as providing the same information for matched residents without diabetes. The final dataset was used to calculate the Charlson Comorbidity Index (CCI) [33] which includes a history of (MI), heart failure (HF), PAD, cerebrovascular disease, chronic pulmonary disease, rheumatic disease, peptic ulcer disease, hemiparesis or paraparesis, renal disease, liver disease, and cancer. For the purposes of the present study, we excluded those conditions coded as diabetes-specific complications (ICD-9-CM 250 and ICD-10-AM E10-14 codes) in FDS participants.

### 2.4. Statistical Analysis

The computer packages IBM SPSS Statistics 25 (IBM Corporation, Armonk, NY, USA) and StataSE 15 (College Station, TX, USA: StataCorp LP) were used for statistical analysis. Data are presented as proportions, mean ± SD, geometric mean (SD range), or, in the case of variables which did not conform to a normal or log-normal distribution, median and inter-quartile range (IQR). Two-sample comparisons were by Fisher’s exact test for proportions, Student’s *t*-test for normally distributed variables, and Mann–Whitney U-test for other variables. Five-year IRs for RRT were derived for each of the four groups defined by type 2 diabetes status and FDS Phase. Incident rate ratios (IRRs) for RRT were then calculated for (i) those with type 2 diabetes in FDS2 versus FDS1, (ii) those without diabetes in FDS2 versus FDS1, and (iii) for those with type 2 diabetes versus no diabetes in FDS1 and FDS2 separately, with incident rate differences (IRDs) also calculated. To allow for differences in age, sex, comorbidities, and management changes between phases, we adjusted for (i) age as the timeline in a Cox model of time to first event for each outcome using people without diabetes in FDS2 as reference, and (ii) in addition, sex, CCI and time between recruitment of each participant and recruitment of the first participant in each phase.

For the pooled FDS1 and FDS2 type 2 diabetes cohort, Cox proportional hazards modeling with backward conditional variable entry (*p* < 0.05) and removal (*p* ≥ 0.05) were used to determine independent predictors of the first episode of RRT during follow-up from clinically plausible baseline variables with *p* < 0.20 in bivariable analyses. To assess the effect of FDS phase, participation in FDS2 versus FDS1 was then added to each most parsimonious model. Missing values were multiply imputed (×20; see Appendix A), defining imputation models that included each outcome.

## 3. Results

### 3.1. Participant Characteristics

The total sample of FDS1 and FDS2 participants combined with the two matched cohorts without diabetes (*n* = 13,995) had a mean ± SD age of 64.8 ± 11.5 years, and 50.4% were males. The baseline characteristics of the four cohorts are summarized in Table 1. Thirty-one (0.2%) participants had chronic renal failure requiring hospitalization for/with RRT at study entry and were therefore excluded from analyses of incident disease. The FDS2 type 2 diabetes cohort had the largest baseline RRT prevalence (0.8%), which was significantly higher than for the cohorts without diabetes (FDS2 0.1% and FDS1 0.2%), but not the FDS1 type 2 after Bonferroni correction for multiple comparisons.

### 3.2. Incident RRT by Type 2 Diabetes Status and FDS Phase

For the type 2 diabetes cohorts, the 5-year IR for RRT was nearly threefold higher in FDS2 compared with FDS1 (IRR (95% CI): 2.85 (1.01–9.87); see Table 2 and Figure 1). Sixteen more people per 10,000 person-years received RRT in FDS2 than FDS1. For the two cohorts without diabetes matched to participants in FDS1 and FDS2, the 5-year IR for RRT was sixfold higher in FDS2 than FDS1 (see Table 2 and Figure 1), but the IRR had a wide confidence interval which spanned unity due to the low number of events. Two more people per 10,000 person-years received RRT in the FDS2 cohort without diabetes than the FDS1 cohort without diabetes.

Excluding participants with an Aboriginal background and their matched counterparts in FDS1, ≤5 (≤0.4%) of the 1271 non-Aboriginal participants with type 2 diabetes started RRT in the five years following study entry compared with ≤5 (≤0.01%) of 5075 in the corresponding matched cohort. The respective figures for FDS2 were 9 (0.6%) of the 1397 non-Aboriginal participants with type 2 diabetes versus 7 (0.1%) of the 5603 in the matched cohort. For the type 2 diabetes cohorts excluding Aboriginal participants, the 5-year IR for RRT was a non-significant 59% higher in FDS2 compared with FDS1 (IRR (95% CI): 1.59 (0.48–6.05); see Table 2 and Figure 1).

The 5-year IRRs and IRDs for RRT by FDS phase for people with type 2 diabetes versus those without diabetes are shown in Table 3 and Figure 2. The 5-year IRRs in FDS2 were less than half of those in FDS1 regardless of whether the whole sample or non-Aboriginal participants were analyzed (9.74 versus 20.5 and 5.13 versus 20.4, respectively). The 5-year IRDs per 10,000 person-years approximately doubled between FDS phases in the total sample but the difference was attenuated in those of non-Aboriginal ethnic background (see Table 3 and Figure 2).

### 3.3. Determinants of Incident Renal Replacement Therapy

The adjusted Cox models for incident RRT are shown in Table 4. In the Cox model which included adjustment for age as the timeline, and using the FDS2 participants without diabetes as the reference group, the highest hazard ratio (HR) for RRT was in the FDS2 participants with type 2 diabetes (10.1), followed by FDS1 participants with type 2 diabetes (3.17). The FDS1 cohort without diabetes were less likely to undergo RRT than the FDS2 cohort without diabetes (0.16) but this did not reach statistical significance in the context of low event numbers. Further adjustment for sex, CCI, and time from start of the respective phase modestly attenuated these findings, as did allowing for the competing risk of death (see Table 4).

Excluding participants with an Aboriginal background and their matched counterparts, the highest hazard ratios (HRs (95% CIs)) for RRT in the Cox model which included adjustment for age as the timeline with the FDS2 no diabetes cohort as reference were in FDS2 participants with type 2 diabetes (5.12), followed by FDS1 participants with type 2 diabetes (3.06) (see Table 4). The FDS1 cohort without diabetes was less likely to undergo RRT than the FDS2 cohort without diabetes (0.15). Further adjustment for sex, CCI, and time from start of the respective phase modestly attenuated these findings, as did allowing for the competing risk of death (see Table 4).

The age of first occurrence of RRT showed no significant difference by phase or type 2 diabetes status, but the oldest age of starting RRT in the 5 years following study entry was 69.9 years for FDS1 and more than 15 years greater at 85.4 years for FDS2. Excluding the eight participants with an Aboriginal background and their matched counterparts increased the mean age of RRT onset for those with type 2 diabetes in FDS2 from 61.6 years (range 33.2–85.4 years) to 74.5 years (range 63.2–85.4 years).

### 3.4. Predictors of Outcomes in Pooled FDS1 and FDS2 Type 2 Diabetes Datasets

At study entry, the pooled FDS1 and FDS2 participants with type 2 diabetes (*n* = 2805) had a mean ± SD age of 64.8 ± 11.5 years; 50.3% were male, and their median (IQR) diabetes duration was 5.0 (1.8–13.0) years. The baseline characteristics of the FDS1 and FDS2 type 2 diabetes cohorts are summarized in Table 5. Compared with the FDS1 cohort, those in FDS2 were older at entry, had greater ethnic heterogeneity, were more fluent in English, and were better educated. They consumed more alcohol but were less likely to be current smokers. They were diagnosed at a younger age and had a longer diabetes duration, their diabetes was more intensively managed pharmacologically, and they had better glycemic control if more self-reported hypoglycemia. The FDS2 participants were more likely to be obese, but had lower systolic blood pressure and more favorable serum lipid levels in association more intensive CVD risk factor pharmacotherapy including use of angiotensin converting enzyme inhibitors and angiotensin receptor blockers. They had lower urinary ACRs and higher eGFRs. More had a past history of stroke, PSN, and RRT, but fewer had PAD and depressive symptoms.

Excluding those with prior RRT, the baseline characteristics of the pooled cohort by incident RRT status are summarized in Appendix A. The most parsimonious Cox models of time to RRT, which utilized age as the time scale and incorporated imputation of missing data (see Appendix A), are shown in Table 6. The predictors of RRT were consistent with recognized risk factors, including urinary ACR, baseline renal impairment, and Aboriginal Australian background. Addition of FDS Phase to the most parsimonious models showed participation in FDS2 versus FDS1 was associated with a non-significant threefold increase in risk of RRT.

After excluding participants with an Aboriginal ethnic background, the only predictors of RRT were urinary ACR and baseline renal impairment (see Table 6). Addition of FDS Phase to the most parsimonious model showed that participation in FDS2 versus FDS1 was associated with a non-significant doubling of the risk of RRT.

## 4. Discussion

The present data show that the incidence of RRT tripled in the 15 years between FDS Phases in people with type 2 diabetes from a representative urban Australian setting. The observation that the IRR was attenuated when Aboriginal participants were excluded from the analysis suggests that rates of RRT have been increasing disproportionately in Aboriginal Australians with type 2 diabetes. Although constrained by a relatively small numbers of events, there was also a non-significant temporal trend to increasing incidence of RRT among matched people without diabetes from the same catchment area. Type 2 diabetes increased the risk of RRT at least 5-fold compared to that in people without diabetes. This increased risk was much greater in the Aboriginal participants, and there was evidence that this group was relatively young at the time RRT was started and that they were more prone to rapid progression to RRT. Multivariable analysis using the combined FDS cohorts confirmed albuminuria as a strong independent risk factor for RRT.

The temporal increase in RRT in type 2 diabetes in the present study is in apparent contrast to the significant reduction in major cardiovascular events found in a similar analysis of FDS2 compared with FDS1 [1]. Nevertheless, an Australian national database study conducted over a shorter time period than FDS (between 2002 and 2013) also found an increasing incidence of ESKD in type 2 diabetes [14], with an overall crude incidence rate similar to that in FDS1 (9.1 versus 8.4/10,000 person-years). The substantial increase in RRT in FDS2 (to 23.9/10,000 person-years) might represent the effect of an additional 3 years of follow-up in FDS2 coupled with progressively greater willingness to treat diabetes-related ESKD in Australia [34]. In addition, 7.1% of FDS2 participants were Aboriginal compared with only 1.5% in FDS1 and 2.1% in the national sample, the latter having acknowledged deficiencies in Aboriginal registrations [14]. Notwithstanding these differences, the national database study found that Aboriginal ESKD rates were four times those in non-Aboriginal registrants [14], a figure similar to the hazards ratios in our Cox proportional hazards models.

There is evidence that improved CVD risk factor management has contributed to reduced macrovascular complications of type 2 diabetes over the past few decades [7,8,9], including in the context of diabetic kidney disease [35]. Better cardiometabolic control, if appropriately implemented across all levels of care [36], might also be expected to slow progression to ESKD [37]. However, increased survival from major CVD events might also increase the number of people with type 2 diabetes requiring RRT. It is of interest that adjusting for the competing risk of premature death did not attenuate the HRs in the present analyses. Although the age at which RRT was started was similar for people with type 2 diabetes in FDS1 and FDS2 (at around 62 years), the SD for this variable in FDS2 was approaching twice that in FDS1 (16 years versus 9 years), suggesting a much broader age spread in the later phase of the FDS. This might mean that, as well as people with type 2 diabetes in FDS2 living longer and being more readily accepted into RRT programs than in FDS1 [34], there are also increasing numbers of younger individuals requiring RRT. The greater age at RRT initiation in non-Aboriginal FDS2 participants with type 2 diabetes (75 years) suggests strongly that Aboriginals with type 2 diabetes are progressing to ESKD at a relatively young age, in accord with previous studies [38]. Consistent with our data, the national database study found that the increasing annual incidence of ESKD was driven by increases among those younger than 50 and older than 80 years [14]. We [39] and others [40] have found that type 2 diabetes in young people is associated with a more severe phenotype and relatively high rates of chronic complications which likely include ESKD.

As well as confirmation of albuminuria as a strong predictive risk factor for ESKD [41], there was indirect evidence from our comparative Cox models that Aboriginals can progress more rapidly from Stage 3 CKD to requiring RRT. In the total sample of FDS1 and FDS2 participants with type 2 diabetes, there was a graded relationship between the baseline eGFR and incident RRT from a HR of 22.4 in those with an eGFR at study entry between 45 and 59 mL/min/1.73 m² to one of 221 for those with an eGFR < 30 mL/min/1.73 m² (Table 6). When the same analysis was repeated in the non-Aboriginal participants, only those with an eGFR < 30 mL/min/1.73 m² were at increased risk (HR 22.8). Consistent with this observation, Aboriginal participants in both FDS phases had worse baseline renal disease risk factor profiles, including a higher HbA1c and a greater proportion of smokers, than their non-Aboriginal counterparts [28].

The present study had limitations. Although the FDS2 cohort was from a typical urban Australian population base and was representative of those with the disease in the study catchment area, the number of RRT endpoints was relatively small and a function of the fixed sample sizes. Nevertheless, the incident rates for RRT and their trends were similar to those in the national sample [14], and the non-significant increasing incidence of RRT among matched people without diabetes was consistent with upward trends in RRT for other causes of ESKD in the Australian general population [34]. Although we had robust identification of Aboriginality in the FDS cohorts, this was not the case for the matched cases without diabetes from the catchment area and so we assumed that those matched with the Aboriginals in the FDS T2D cohorts were also Aboriginal. Given that the average age of Aboriginals in Australia is 20 years compared with 38 years in the non-Aboriginal population [42], it is likely that the groups of individuals without diabetes matched with the relatively young Aboriginals in FDS1 and FDS2 were enriched with Aboriginals. The uptake of the glucagon-like peptide 1 receptor agonists and sodium-glucose co-transporter-2 inhibitors (SGLT2i), classes of blood glucose-lowering therapies with glucose-independent benefits for renal disease, especially SGLT2i [43], was relatively low in the FDS2 cohort (<5% in each case at the end of the follow-up period) given that they were first introduced into Australia when the study was in progress. In any case, any effect they may have had on the incidence of RRT would have been an attenuation of the increase between FDS1 and FDS2. The strengths of the study include the detailed participant-level data and access to a validated data linkage system with ascertainment of relatively hard renal-related endpoints.

In conclusion, the present study provides patient-level evidence from a representative urban community sample that the incidence of RRT in type 2 diabetes has increased over the last few decades. This finding aligns with recent national Australian administrative data [14], but we provide additional evidence of the increased risk of ESKD amongst Aboriginal people with type 2 diabetes who can present at a young age and can progress rapidly to RRT in the presence of relatively adverse renal risk factors. The increasing incidence of RRT in type 2 diabetes may be multifactorial, but increased willingness to implement this management strategy in older individuals with comorbidities [34] as well as a more severe clinical phenotype in younger, especially Aboriginal patients, appear largely responsible.

## Figures and Tables

**Figure 1 jcm-11-00695-f001:**
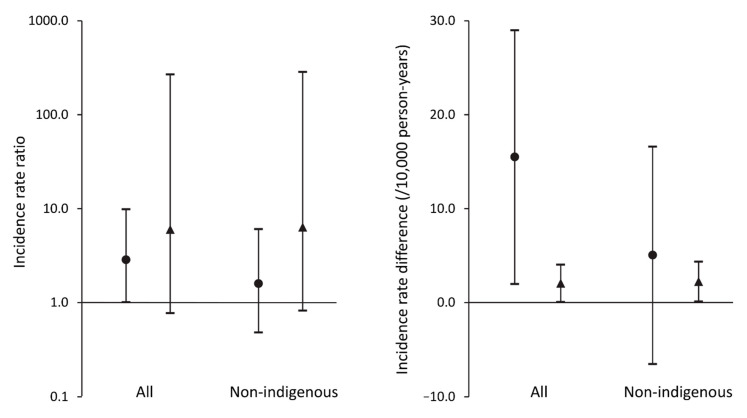
Incidence rate ratios (IRRs) and incidence rate differences (IRDs) for renal replacement therapy (RRT) in participants from the Fremantle Diabetes Study Phase 1 (FDS1) versus those in participants from Phase 2 (FDS2) with type 2 diabetes (●) and without known diabetes (▲) from the FDS catchment area. Data are IRR (left panel) and IRD (right panel) with 95% confidence intervals (vertical bars) for all participants and those who were not of Aboriginal ethnic origin.

**Figure 2 jcm-11-00695-f002:**
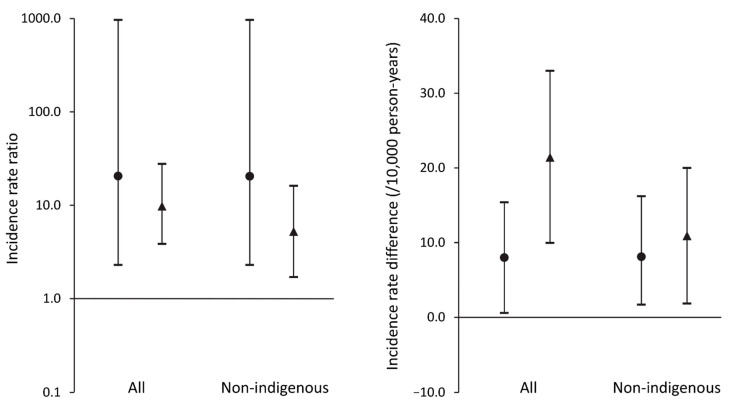
Incidence rate ratios (IRRs) and incidence rate differences (IRDs) for renal replacement therapy (RRT) in participants from the Fremantle Diabetes Study Phase 1 (FDS1); (●) with versus without type 2 diabetes and in participants from Phase 2 (FDS2); (▲) with versus without type 2 diabetes. Data are IRR (left panel) and IRD (right panel) with 95% confidence intervals (vertical bars) for all participants and those who were not of Aboriginal ethnic origin.

**Table 1 jcm-11-00695-t001:** Characteristics at study entry of type 2 diabetes FDS1 and FDS2 participants and their matched cohorts.

	FDS1 Type 2 Diabetes	FDS1 No Diabetes	FDS2 Type 2 Diabetes	FDS2 No Diabetes	*p*-Value
Number (%)	1291	5159	1509	6036	
Age at FDS entry (years)	64.0 ± 11.2	64.0 ± 11.2	65.4 ± 11.7 **^,†††^	65.4 ± 11.7 ***^,†††^	<0.001
Sex (% male)	48.7	48.7	51.8	51.8 ^††^	0.005
Aboriginal Australian (%)	1.5	-	7.1	-	-
History of hospitalization for/with RRT ^a^ (%)	0.2	0.2	0.8 ^††^	0.1 ^‡‡‡^	<0.001
Charlson Comorbidity Index ^b^ (%)		***	**^,^^†††^	***^,‡‡‡^	<0.001
0	71.6	85.6	75.1	86.5	
1–2	22.0	11.1	16.8	9.8	
≥3	6.4	3.3	8.0	3.7	

^a^ RRT = renal replacement therapy; ^b^ in the last 5 years, excluding diabetes and its complications; ** *p* < 0.01, *** *p* < 0.001 vs. FDS1 type 2 diabetes; ^††^ *p* < 0.01, ^†††^ *p* < 0.001 vs. FDS1 no diabetes; ^‡‡‡^ *p* < 0.001 vs. FDS2 type 2 diabetes, Bonferroni-corrected for multiple comparisons.

**Table 2 jcm-11-00695-t002:** Five-year incidence rates (IR; per 10,000 person-years), incidence rate ratios (IRR; 95% CI) and incident rate differences (IRD; per 10,000 person-years) for first hospitalization for/with renal replacement therapy in FDS2 versus FDS1 type 2 diabetes participants and their matched counterparts without diabetes for the total sample (above) and non-Aboriginal participants (below).

		FDS1			FDS2		FDS2:FDS1	FDS2—FDS1
Type 2 Diabetes	N	Follow-Up (Years)	IR	N	Follow-Up (Years)	IR	IRR (95% CI)	IRD (95% CI)
Total sample								
Yes	≤5 *	5956	8.40 (2.73–19.6)	17	7115	23.9 (13.9–38.3)	2.85 (1.01–9.87)	15.5 (1.97–29.0)
No	≤5 *	24,376	0.41 (0.01–2.29)	7	28,540	2.45 (0.99–5.05)	5.98 (0.77–269)	2.04 (0.06–4.03)
Non-Aboriginal participants								
Yes	≤5 *	5872	8.52 (2.76–19.9)	9	6636	13.6 (6.20–25.7)	1.59 (0.48–6.05)	5.05 (−6.54–16.6)
No	≤5 *	24,009	0.42 (0.01–2.32)	7	26,467	2.64 (1.06–5.45)	6.35 (0.82–286)	2.23 (0.11–4.35)

* Actual numbers not given to preserve confidentiality.

**Table 3 jcm-11-00695-t003:** Five-year incidence rate ratios (IRR; 95% CI) and incident rate differences (IRD; per 10,000 person-years) for first hospitalization for/with renal replacement therapy in type 2 diabetes participants versus their matched counterparts without diabetes in FDS1 and FDS2 for the total sample (above) and non-Aboriginal participants (below).

	IRR (95% CI)		IRD (95% CI)	
	FDS1	FDS2	FDS1	FDS2
Total sample	20.5 (2.29–968)	9.74 (3.84–27.8)	7.98 (0.58–15.4)	21.4 (9.94–33.0)
Non-Aboriginal participants	20.4 (2.29–967)	5.13 (1.70–16.20)	8.10 (0.59–15.6)	10.9 (1.84–20.0)

**Table 4 jcm-11-00695-t004:** Cox and Fine and Gray models, and age at first hospitalization for/with renal replacement therapy (RRT), occurring within 5-years of study entry by FDS Phase and type 2 diabetes status in those with no prior hospitalization for/with RRT.

Phase	Type 2 Diabetes	N	Events	CsHR ^a^ (95% CI)	CsHR ^b^ (95% CI)	SdHR ^a^ (95% CI)	SdHR ^b^ (95% CI)	Age at Event (Years)
Total sample								
2	No	6027	7	1.0	1.0	1.0	1.0	67.9 ± 7.6
2	Yes	1497	17	10.1 (4.20, 24.5)	7.17 (2.90, 17.7)	9.89 (4.08, 24.0)	7.25 (2.88, 18.2)	61.6 ± 16.0
1	No	5151	≤5 *	0.16 (0.02, 1.28)	0.16 (0.02, 1.32)	0.15 (0.02, 1.24)	0.15 (0.02, 1.26)	49.4
1	Yes	1289	≤5 *	3.17 (1.01, 10.0)	2.22 (0.69, 7.27)	2.97 (0.95, 9.33)	2.11 (0.58, 7.65)	62.9 ± 8.9
Non-Aboriginal participants								
2	No	5603	7	1.0	1.0	1.0	1.0	67.9 ± 7.6 (55–75)
2	Yes	1397	9	5.12 (1.91, 13.8)	3.69 (1.35, 10.1)	5.03 (1.88, 13.5)	3.86 (1.44, 10.4)	74.5 ± 6.4 (63–85)
1	No	5075	≤5 *	0.15 (0.02, 1.22)	0.14 (0.02, 1.17)	0.14 (0.02, 1.15)	0.13 (0.02, 1.11)	49.4
1	Yes	1271	≤5 *	3.06 (0.97, 9.67)	1.96 (0.60, 6.37)	2.86 (0.92, 8.89)	1.86 (0.53, 6.53)	62.9 ± 8.9 (48–69)

Cs = cause-specific; sd = subdistribution; HR = hazard ratio; CI = confidence interval; * actual numbers not given to preserve confidentiality; ^a^ adjusted for age as timeline; ^b^ adjusted for age as timeline, sex, Charlson’s Comorbidity Index, time from recruitment of first participant in each Phase to study entry for each participant/matched.

**Table 5 jcm-11-00695-t005:** Comparison of the characteristics of type 2 diabetes participants in FDS1 and FDS2 at study entry.

	FDS1	FDS2	*p*-Value
Number (%)	1296 (46.2)	1509 (53.8)	
Time from start of Phase to participant entry (years)	1.21 ± 0.83	1.59 ± 0.93	<0.001
Age at FDS entry (years)	64.0 ± 11.3	65.4 ± 11.7	0.001
Sex (% male)	48.6	51.8	0.10
Overseas born (%)	46.8	43.5	0.09
Ethnic background (%): Anglo-Celt	61.4	52.6	
Southern European	17.7	12.9	
Other European	8.5	7.4	<0.001
Asian	3.4	4.3	
Aboriginal Australian	1.5	7.1	
Mixed/other	7.5	15.8	
Not fluent in English (%)	15.3	10.8	<0.001
Education beyond primary level (%)	74.0	86.8	<0.001
Currently married/de facto (%)	65.7	62.7	0.11
Alcohol (standard drinks/day)	0 (0–0.8)	0.1 (0–1.2)	<0.001
Smoking status (%) Never	44.7	45.5	
Ex-	40.2	43.9	0.001
Current	15.1	10.7	
Age at diagnosis (years)	57.9 ± 11.7	55.6 ± 12.4	<0.001
Duration of diabetes (years)	4.0 (1.0–9.0)	8.0 (2.7–15.4)	<0.001
Diabetes treatment (%): Diet	31.9	24.6	
Oral agents	55.7	53.4	<0.001
Insulin ± oral agents	12.3	22.0	
Fasting serum glucose (mmol/L)	8.0 (6.5–10.3)	7.2 (6.2–8.9)	<0.001
HbA_1c_ (%)	7.2 (6.2–8.5)	6.8 (6.2–7.7)	<0.001
HbA_1c_ (mmol/mol)	55 (44–69)	51 (44–61)	<0.001
Self-reported hypoglycemia last year (%):	22.9	33.9	<0.001
Body mass index (kg/m^2^)	29.6 ± 5.4	31.3 ± 6.1	<0.001
Obesity (% by waist circumference)	64.5	70.9	<0.001
Antihypertensive medication (%)	50.9	73.2	<0.001
Angiotensin converting enzyme inhibitors/angiotensin receptor antagonists (%)	21.8	64.5	<0.001
Systolic blood pressure (mm Hg)	151 ± 24	146 ± 22	<0.001
Diastolic blood pressure (mm Hg)	80 ± 11	80 ± 12	0.55
Heart rate (/min)	70 ± 12	70 ± 12	0.85
Lipid-modifying medication (%)	10.5	68.2	<0.001
Total serum cholesterol (mmol/L)	5.5 ± 1.1	4.4 ± 1.1	<0.001
Serum HDL-cholesterol (mmol/L)	1.06 ± 0.33	1.24 ± 0.34	<0.001
Serum triglycerides (mmol/L)	2.2 (1.2–3.9)	1.5 (0.9–2.5)	<0.001
Serum uric acid (mmol/L)	0.38 ± 0.11	0.34 ± 0.09	<0.001
Aspirin use (%)	22.0	37.2	<0.001
Urinary albumin:creatinine (mg/mmol)	5.2 (1.5–17.8)	3.3 (0.8–12.7)	<0.001
eGFR categories (%): ≥90 mL/min/1.73 m^2^	32.2	38.9	
60–89 mL/min/1.73 m^2^	49.8	44.6	
45–59 mL/min/1.73 m^2^	11.9	8.8	<0.001
30–44 mL/min/1.73 m^2^	4.4	4.9	
<30 mL/min/1.73 m^2^	1.7	2.8	
Hospitalization for/with RRT (%)	0.2	0.8	0.016
Atrial fibrillation (%)	4.9	4.6	0.72
Hospitalization for/with heart failure (%)	8.3	6.4	0.07
Hospitalization for/with myocardial infarction (%)	8.7	8.1	0.59
Ischemic heart disease (%)	29.6	28.8	0.68
Hospitalization for/with stroke (%)	0.4	3.0	<0.001
Cerebrovascular disease (%)	10.0	11.2	0.30
Hospitalization for lower extremity amputation (%)	1.2	1.1	0.86
Peripheral arterial disease (%)	29.3	22.6	<0.001
Peripheral sensory neuropathy (%)	30.8	58.2	<0.001
Depressive symptoms (%)	31.5	23.1	<0.001
ApoE genotype (%): 22	0.8	0.4	
23	11.8	11.4	
24	2.3	2.5	0.54
33	65.5	64.5	
34	18.3	19.4	
44	1.2	1.8	
ApoE4 allele (%)	21.9	23.7	0.27
Charlson Comorbidity Index ^a^ (%):			0.001
0	71.5	75.1	
1–2	22.1	16.8	
≥3	6.4	8.0	

^a^ In the last 5 years, excluding diabetes and its complications.

**Table 6 jcm-11-00695-t006:** Multiply imputed Cox models of time to first hospitalization for/with RRT, with age as time scale in pooled FDS1 and FDS2 participants with type 2 diabetes (HR (95% CI)). Model 1: Most parsimonious model; Model 2: Model 1 with FDS2 added.

	Model 1	Model 2
Total sample		
*n*/N	22/2787	22/2787
Aboriginal	6.41 (1.02–40.1)	3.84 (0.56–26.3)
Ln(ACR (mg/mmol)) *	2.52 (1.77–3.58)	2.57 (1.81–3.65)
eGFR (CKD-EPI):		
45–59 mL/min/1.73 m^2^	22.4 (2.34–214)	22.1 (2.38–205)
30–44 mL/min/1.73 m^2^	39.3 (4.03–383)	47.1 (4.59–482)
<30 mL/min/1.73 m^2^	221 (23.4–2081)	230 (26.1–2036)
FDS2		3.13 (0.81–12.1)
Non-Aboriginal participants		
*n*/N	14/2673	14/2673
Ln(ACR (mg/mmol)) *	2.78 (1.84–4.20)	2.78 (1.85–4.17)
eGFR (CKD-EPI) <30 mL/min/1.73 m^2^	22.8 (5.91–87.8)	24.5 (6.42–93.2)
FDS2		2.22 (0.59–8.35)

* An increase of 1 in ln(ACR (mg/mmol)) equates to an increase of 2.72 in ACR (mg/mmol).

## Data Availability

The datasets generated during this study and/or as a result of analysis are available from the corresponding author on reasonable request.

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
