# Peer review of "Temporal Trends in Renal Replacement Therapy in Community-Based People with or without Type 2 Diabetes: The Fremantle Diabetes Study"

_jcm, 2022, doi:10.3390/jcm11030695_

Round 1

Reviewer 1 Report

The paper is interesting, quite original and well written. The conclusions are supported by results. Tables and figures are clear.

However, this reviewer raises some issues that the authors have to address..

1- No information is provided by the authors on the drug therapy of patients. For example, drugs active on RAS and, more recently, SGLT2i are known to modify the progression of CKD towards ESRD. Recently, the nephroprotective effect of gliflozins has also been demonstrated in non-diabetic subjects. The lack of drug history of the patients represents a major limitation of the study that the authors have to address in the discussion.

2- In Conclusions, the authors state “The increasing incidence of RRT in type 2 diabetes may be multifactorial, but increased willingness to implement this management strategy… ap-pear largely responsible.”. This hope has found application for the first time very recently in a multicenter randomized trial in the NID-2 study in which subjects with diabetic kidney disease treated with intensified multifactorial therapy showed a significant and lasting reduction in mortality and MACE (Cardiovasc Diabetol (2021) 20:145. doi: 10.1186/s12933-021-01343-1). This very important issue should be addressed in discussion.

3- Management of risk factors in subjects with advanced nephropathy is a very hot issue. These patients, despite the worst renal and cardiovascular prognosis, are at high risk of being under-treated independently of the type of clinical setting. (J Hypertens. 2006 Aug;24(8):1655-61. doi: 10.1097/01.hjh.0000239303.93872.31.). This issue and above reference should be added in discussion.

4- Authors observed that Type 2 diabetes increased risk of RRT at least 5-fold compared to that in people without diabetes, and multivariable confirmed albuminuria as a strong independent risk factor for RRT. Moreover, it is known that albuminuria has a relevant prognostic effect on CV morbidity and mortality. Notably, this effect is especially pronounced when GFR is normal or near normal. (Nephrol Dial Transplant. 2012 Jun;27(6):2269-74. doi: 10.1093/ndt/gfr644. Epub 2011 Nov 16.). This issue is worthy of a comment in discussion.

Author Response

1- No information is provided by the authors on the drug therapy of patients. For example, drugs active on RAS and, more recently, SGLT2i are known to modify the progression of CKD towards ESRD. Recently, the nephroprotective effect of gliflozins has also been demonstrated in non-diabetic subjects. The lack of drug history of the patients represents a major limitation of the study that the authors have to address in the discussion.

Response: We had detailed data on medications which is summarised in the Table 5 which contains patient characteristics. We have added to these data by including the RAs blocker category of antihypertensive therapy which was consistent with the difference between FDS phase in the antihyptenstive category as a whole. This is now also detailed in the Results under subheading ‘Predictors of outcomes in pooled FDS1 and FDS2 type 2 diabetes datasets” on page 11 of the revised manuscript. In addition we have included mention of the low level gliflozin use in the Discussion on page 14: “The uptake of the glucagon-like peptide 1 receptor agonists and sodium-glucose co-transporter-2 inhibitors (SGLT2i), classes of blood glucose-lowering therapies with glucose-independent benefits for renal disease especially SGLT2i [43], was relatively low in the FDS2 cohort (<5% in each case at the end of the follow-up period) given that they were first introduced into Australia when the study was in progress. In any case, any effect they may have had on the incidence of RRT would have been an attenuation of the increase between FDS1 and FDS2.” 

2- In Conclusions, the authors state “The increasing incidence of RRT in type 2 diabetes may be multifactorial, but increased willingness to implement this management strategy… ap-pear largely responsible.”. This hope has found application for the first time very recently in a multicenter randomized trial in the NID-2 study in which subjects with diabetic kidney disease treated with intensified multifactorial therapy showed a significant and lasting reduction in mortality and MACE (Cardiovasc Diabetol (2021) 20:145. doi: 10.1186/s12933-021-01343-1). This very important issue should be addressed in discussion.

Response: We have included the suggested reference in the Discussion on page 13: “There is evidence that improved CVD risk factor management has contributed to reduced macrovascular complications of type 2 diabetes over the past few decades [7-9], including in the context of diabetic kidney disease [35].”

3- Management of risk factors in subjects with advanced nephropathy is a very hot issue. These patients, despite the worst renal and cardiovascular prognosis, are at high risk of being under-treated independently of the type of clinical setting. (J Hypertens. 2006 Aug;24(8):1655-61. doi: 10.1097/01.hjh.0000239303.93872.31.). This issue and above reference should be added in discussion.

Response: We have included the suggested reference in the Discussion on page 13: “Better cardiometabolic control, if appropriately implemented across all levels of care [36], might also be expected to slow progression to ESKD [37].”

4- Authors observed that Type 2 diabetes increased risk of RRT at least 5-fold compared to that in people without diabetes, and multivariable confirmed albuminuria as a strong independent risk factor for RRT. Moreover, it is known that albuminuria has a relevant prognostic effect on CV morbidity and mortality. Notably, this effect is especially pronounced when GFR is normal or near normal. (Nephrol Dial Transplant. 2012 Jun;27(6):2269-74. doi: 10.1093/ndt/gfr644. Epub 2011 Nov 16.). This issue is worthy of a comment in discussion.

Response: The relationship between albuminuria and CVD morbidity and mortality is not relevant to this paper as renal replacement therapy is the endpoint of interest and so we have not added the suggested reference.

Reviewer 2 Report

This is an excellent paper scientifically, as well as linguistically, regarding the need for renal replacement therapies in diabetic patients in Australia.
The discussion is comprehensive and thorough and could be supplemented, if necessary, to the effect that dialysis has been much more liberal in the two decades that lay between the two comparison areas; regardless of diabetes, the age of dialysis patients has also risen accordingly. In line with this, more or less all socioeconomic and clinical criteria improved in the two comparison periods, only the extent of diabetic neuropathy was now much more frequent (which indicates a much more extensive and attentive diagnosis). 
Apart from this possible extension of the discussion, this work does not need any further optimization in my opinion. 

Author Response

The discussion is comprehensive and thorough and could be supplemented, if necessary, to the effect that dialysis has been much more liberal in the two decades that lay between the two comparison areas; regardless of diabetes, the age of dialysis patients has also risen accordingly. In line with this, more or less all socioeconomic and clinical criteria improved in the two comparison periods, only the extent of diabetic neuropathy was now much more frequent (which indicates a much more extensive and attentive diagnosis).  Apart from this possible extension of the discussion, this work does not need any further optimization in my opinion. 

Response: We have altered the Discussion on page 14 to emphasise the temporal increase in RRT outside the context of type 2 diabetes: “Nevertheless, the incident rates for RRT and their trends were similar to those in the national sample [14], and the non-significant temporal trend to an increasing incidence of RRT among matched people without diabetes was consistent with trends for other causes of ESKD in the Australian general population [34].”

Round 2

Reviewer 1 Report

The authors addressed the issues raised by this reviewer. No further comments.